# Self-Sampled Gargle Water Direct RT-LAMP as a Screening Method for the Detection of SARS-CoV-2 Infections

**DOI:** 10.3390/diagnostics12040775

**Published:** 2022-03-22

**Authors:** Skaiste Arbaciauskaite, Pouya Babakhani, Natalia Sandetskaya, Dalius Vitkus, Ligita Jancoriene, Dovile Karosiene, Dovile Karciauskaite, Birute Zablockiene, Dirk Kuhlmeier

**Affiliations:** 1Department of Diagnostics, Fraunhofer Institute for Cell Therapy and Immunology IZI, Perlickstraße 1, 04103 Leipzig, Germany; natalia.sandetskaya@izi.fraunhofer.de (N.S.); dirk.kuhlmeier@izi.fraunhofer.de (D.K.); 2Institute of Cell Biology and Neurobiology, Charité—Universitätsmedizin Berlin, Charitéplatz 1, 10117 Berlin, Germany; 3Department of Computer Science, University of Bath, Claverton Down, Bath BA2 7AY, UK; pb603@bath.ac.uk; 4Institute of Biomedical Sciences, Vilnius University Faculty of Medicine, M.K. Ciurlionio 21, LT-03101 Vilnius, Lithuania; dalius.vitkus@santa.lt (D.V.); dovile.karciauskaite@mf.vu.lt (D.K.); 5Centre of Laboratory Medicine, Vilnius University Hospital Santaros Klinikos, Santariskiu 14, LT-08406 Vilnius, Lithuania; dovile.karosiene@santa.lt; 6Clinic of Infectious Diseases and Dermatovenerology, Institute of Clinical Medicine, Vilnius University Faculty of Medicine, M.K. Ciurlionio 21, LT-03101 Vilnius, Lithuania; ligita.jancoriene@santa.lt (L.J.); birute.zablockiene@santa.lt (B.Z.); 7Center of Infectious Diseases, Vilnius University Hospital Santaros Klinikos, Santariskiu 14, LT-08406 Vilnius, Lithuania

**Keywords:** loop-mediated isothermal amplification, self-sampling, gargle water, SARS-CoV-2, COVID-19, viral diseases, LAMP, direct RT-LAMP, RT-PCR

## Abstract

We assessed the viability of self-sampled gargle water direct RT-LAMP (LAMP) for detecting SARS-CoV-2 infections by estimating its sensitivity with respect to the gold standard indirect RT-PCR of paired oro-nasopharyngeal swab samples. We also assessed the impact of symptom onset to test time (STT)—i.e., symptom days at sampling, on LAMP. In addition, we appraised the viability of gargle water self-sampling versus oro-nasopharyngeal swab sampling, by comparing paired indirect RT-PCR results. 202 oro-nasopharyngeal swab and paired self-sampled gargle water samples were collected from hospital patients with COVID-19 associated symptoms. LAMP, indirect and direct RT-PCR were performed on all gargle water samples, and indirect RT-PCR was performed on all oro-nasopharyngeal samples. LAMP presented a sensitivity of 80.8% (95% CI: 70.8–90.8%) for sample pairs with sub-25 Ct oro-nasopharyngeal indirect RT-PCR results, and 77.6% (66.2–89.1%) sensitivity for sub-30 Ct samples with STT ≤ 7 days. STT, independently of Ct value, correlated negatively with LAMP performance. 80.7% agreement was observed between gargle water and oro-nasopharyngeal indirect RT-PCR results. In conclusion, LAMP presents an acceptable sensitivity for low Ct and low STT samples. Gargle water may be considered as a viable sampling method, and LAMP as a screening method, especially for symptomatic persons with low STT values.

## 1. Introduction

The COVID-19 pandemic has incentivized the development of new diagnostic solutions for infectious diseases. Sample processing technologies, e.g., RT-PCR, antigen, and serological testing, have been co-developing alongside different sampling methods, e.g., sputum, gargle water, anal swabs, with the aim of attaining the best possible sensitivity and specificity with minimal time and resources consumption [1,2]. Oro-nasopharyngeal swab indirect RT-PCR (hereafter RT-PCR) remains the gold standard for SARS-CoV-2 detection, even though it is generally expensive, time consuming, and personnel intense [3]. Moreover, oro-nasopharyngeal swab sampling is not well tolerated by many patients. An alternative to RT-PCR is direct reverse transcription loop-mediated isothermal amplification (direct RT-LAMP or just LAMP). A LAMP reaction can be completed in about 30 min, and requires no temperature cycling [4]. Furthermore, LAMP allows for various read-out options, such as fluorescence, electrochemical, or probe-based methods combined with real-time or endpoint detection [5,6,7]; in this study, we opted for colorimetric endpoint read-outs. Due to the clearly visible color changes induced by amplification, a colorimetric readout can be done with the naked eye—with a resultant pink color indicating a negative result and yellow indicating a positive result. Hence, colorimetric read-outs reduce the need for sophisticated equipment, and help make the testing process relatively simple, affordable, and instrument-free [8,9,10,11]. Moreover, comfortable sampling methods for medical staff and patients may be combined with LAMP, as it is compatible with the majority of biospecimens [12,13,14]. For instance, collecting gargle water instead of oro-nasopharyngeal swab samples can reduce the workload for medical staff, and provide patients with a more comfortable sampling procedure [15].

Until now, few studies have been conducted that have specifically evaluated the feasibility of LAMP in a clinical setting, and although spike-in experiments with RNA are informative, differences have been found when comparing such experiments to those using clinical RNA samples isolated from swab specimens [16]. Nevertheless, data published from such trials (with limited sample sizes of mostly oro-nasopharyngeal swabs) suggest very high analytical sensitivities for LAMP, with the limit of detection ranging from at least 100 copies of viral RNA per reaction down to as little as 30 or even 2 copies of viral RNA per reaction [17,18,19,20,21,22]. Kellner et al. and Bockelmann et al. studied the application of indirect RT-LAMP to gargle water, among other sample types like nasopharyngeal swabs and sputum [23,24,25]. In addition to a rapid lysis protocol, the researchers applied an additional bead-based RNA pre-enrichment step, which removed inhibitors and improved the accuracy of detection, but this additional step, of course, incurred additional labor, resource, and time costs. Therefore, in this study, we performed direct LAMP on gargle water samples without any pre-enrichment steps and then compared the colorimetric results to the RT-PCR results obtained from the same gargle water samples, as well as from paired oro-nasopharyngeal swab samples (i.e., the gold standard method), in order to assess the suitability of this method for point-of-care rapid testing. Although LAMP performs better with an RNA extraction step [16], LAMP without RNA extraction presents a more affordable and rapid procedure, and therefore was investigated, as the sensitivity and specificity may still be acceptable for screening applications [7]. In particular, if the direct LAMP procedure can be further optimized, it may present a more sensitive alternative to rapid antigen tests, which have presented unsatisfactory performances in clinical investigations [26,27,28,29].

On a further note, the contagiousness of individuals who have been technically diagnosed as SARS-CoV-2 positive by RT-PCR, but with very late Ct values and/or high symptoms onset to test time (STT) values, remains in question [30,31]. In certain circumstances, e.g., when an infection is known to be have already been present for a long enough period of time, it may be beneficial to avoid “over-diagnosing”; e.g., if it would amount to imposing the unnecessary isolation of non-contagious persons [32,33], or if an outbreak nears a peak and hospital capacities become neigh exhausted, and it becomes valuable to identify people who are not infectious, so as to discharge them. Indeed, it is known that high SARS-CoV-2 RNA loads can be detected, even after infectious viruses are no longer detectable within cell cultures, especially at time periods beyond 7–14 days STT [34,35,36,37]. However, it is still unclear exactly how, with respect to increasing Ct and STT values, the actual degree of infectivity declines [38,39]. Therefore, we traced the performance of gargle water LAMP with respect to a range of different Ct and STT value cut-offs for the clinically derived samples. Furthermore, by exploiting a logistic model, we assessed whether STT, while controlling for sample Ct value (taken as a proxy for viral load), impacted the performance of LAMP.

In an additional section of data analysis, we assessed the sensitivity of gargle water direct RT-PCR against the gold standard oro-nasopharyngeal indirect RT-PCR. We note that in previous studies, direct RT-PCR has been identified as a more resource efficient alternative to indirect RT-PCR testing [40,41,42]. Our assessment of direct RT-PCR was carried out in the same manner as was done for LAMP. In turn, this allowed us to compare the performance of LAMP and direct RT-PCR, both being direct methods performed on the same gargle water samples.

Finally, a brief comparison of gargle water self-sampling against the more established method of oro-nasopharyngeal swab sampling was carried out. A comparison was made over the indirect RT-PCR results derived from paired gargle water and oro-nasopharyngeal swab samples.

## 2. Materials and Methods

### 2.1. Study Design

This study aimed to assess the feasibility of implementing a gargle water LAMP-based rapid test workflow as a diagnostic routine in a pandemic situation. The main objective of this study was to assess the performance (specifically the sensitivity) of the gargle water LAMP test for the detection of SARS-CoV-2 against gold standard RT-PCR tests in a clinical setting. Additionally, all analyses were to be performed with respect to symptoms onset to test time (STT) in order to evaluate the impact of this variable on the test performance.

The study was approved by the relevant institutional review boards (Vilnius Regional Ethics Committee for Biomedical Research). The trial inclusion criteria were as follows: Presenting, during sampling, or having presented upon admission to the hospital, symptoms associated with COVID-19;Being ≥ 18 years of age on the date of sampling;Presenting the ability and willingness to provide informed consent;Presenting willingness to comply with the study procedures.

The required sample size was computed using the formula for sample size in studies with binary outcomes [43]. For the sake of computation, simplifying assumptions were made that the LAMP method would present a sensitivity of 80% (following the minimal acceptable sensitivity as indicated by the WHO’s priority target product profiles for priority COVID-19 diagnostics) and that all recruited patients would have active Sars-CoV-2 infections (i.e., be true positives). Furthermore, a confidence level of 90% and a margin of error of 0.05 were chosen for the calculation. We then derived a sample size of 174 via the following formula: $$n=\lceil \frac{Z_{\alpha /2}^{2}P\left(1-P\right)}{d^{2}}\rceil$$
where *Z_α⁄2_* is the appropriately chosen *Z*-score, *P* is the prior estimate of the sensitivity, and d is the margin of error. So, via this formula, we estimated the need for roughly 174 sample pairs. However, as the underlying assumption that all sampled patients would have active infections would most likely be false, this calculation was regarded as an underestimate, and therefore an additional 28 sample pairs were collected for the trial (yielding a total of 202) following the availability of eligible patients in the hospital during the sample collection period. That is, 202 oro-nasopharyngeal swabs and 202 matched gargle water samples were collected following the inclusion criteria outlined above about present illness and patient consent.

Oro-nasopharyngeal swabs were first collected using a standardized method by a health care professional. One swab was inserted through one nostril parallel to the palate and rotated several times prior to being removed, as per the Center for Disease Control instructions for collection. Another swab was smeared against the posterior oropharynx and the tonsillar arches. The swabs were placed into universal transport medium (UTM; Copan, Brescia, Italy) and then the RT-PCR procedure was promptly performed. The gargle water samples were collected by the participants themselves. Trained healthcare personnel observed and showed participants how to self-collect gargle water. Each participant received a tube with 10 mL of sterile 0.9% saline and had to gargle it for around 5–10 s, and then spit out the contents back into the same tube. The gargle water samples were placed in −20 °C for later processing via indirect and direct RT-PCR, and LAMP. All of the samples were collected and processed in accordance with standard biosafety rules. It should also be noted that the term “indirect” refers to when a testing method was applied after RNA extraction; “direct”, in turns, refers to when there had only been the heat lysis of a sample, as described below for LAMP.

### 2.2. Sample Preparation for LAMP

The gargle water samples were spun down at 350× *g* for 1 min, and 1 mL of the obtained pellet was transferred into a new 1.5 mL vial. Six microliters of the pelleted sample were mixed with 69 µL RNA protectant solution: RNAsecure™ RNase Inactivation Reagent (ThermoFischer Scientific, Darmstadt, Germany) diluted to 1:23 with molecular biology grade water. The sample was incubated for 5 min at 95 °C to inactivate the virus and was cooled down to room temperature. 

### 2.3. Nucleic Acids Extraction for Indirect RT-PCR 

The total nucleic acids were extracted from 200 µL of sample on the King Fisher Flex automated extraction system (Thermo Fisher Scientific, Carlsbad, CA, USA) using the MagMAX Viral/Pathogen Nucleic Acid Isolation Kit (MVP II) (Thermo Fisher Scientific, Carlsbad, CA, USA). The eluate of 50 µL was submitted for a RT-qPCR assay. Then, 5 µL of the sample was added to the RT-PCR reaction. 

### 2.4. SARS-CoV-2 LAMP

The RT-LAMP assay for the detection of SARS-CoV-2 RNA was based on the SARS-CoV-2 Rapid Colorimetric LAMP Assay Kit (New England Biolabs, Frankfurt, Germany). According to the manufacturer, the analytical sensitivity of this kit, at 50 genomic copy equivalents per reaction, is >95% and the specificity is >99%. The reagent mix for the study was prepared in dried format, and pre-dosed for the individual reactions in 0.2 mL reaction vials. Each reaction contained 12.5 µL of WarmStart Colorimetric LAMP 2X Master Mix with UDG, 2.5 µL 10× SARS-CoV-2 LAMP Primer Mix (N/E), 2.5 µL 0.4 M Guanidine Hydrochloride (all listed reagents above are components of the kit), 0.5 µL of 12.5 mg/mL BSA (molecular biology grade, Sigma Aldrich, Taufkirchen, Germany), and 3 µL of 50% trehalose (Sigma Aldrich, Taufkirchen, Germany). The reagents were frozen at −80 °C for 30 min and dried in a lyophilizer (Vaco 2-II, Zirbus technology, Bad Grund, Germany) for 3 h at 3 mbar. According to communications with the manufacturer, the Master Mix contains glycerol that hinders proper freeze-drying. Therefore, the given process should not be viewed as a complete lyophilization, and it did not provide the long-term stability of the reagents: we observed a loss of sensitivity (1 log base 10 of RNA concentration) after 3 weeks of storage at 4 °C (data not shown). Yet the preservation of the reagents in the pre-dosed dried format enabled a convenient test routine in the clinical laboratory. All reagents were used within a period of 2 weeks after drying.

Twenty-five microliters of the inactivated sample were added to a vial with the pre-dosed dried LAMP reagents. LAMP was performed for 30 min at 65 °C. The sample was assessed as positive when the color of the reaction changed from pink to yellow and negative when the color remained pink. The color occasionally turned to an intermediate value of orange, in which case the test was, strictly speaking, inconclusive. However, a color change to orange may be interpreted as a weakly positive result and it is an indication of a possible infection, and so was conservatively classified as a positive result. The results were photographically documented. 

### 2.5. RT-PCR

The RT-PCR was performed with a TaqPath COVID-19 Combo qPCR kit (ThermoFischer Scientific, Waltham, MA, USA). According to the manufacturer, the limit of detection (LoD 95%) of this kit goes down to 10 genomic copy equivalents per reaction (regarding specificity, the kit did not yield any false positives in a clinical evaluation carried out by the manufacturer). The RT-PCR reaction consisted of 12.5 μL nuclease-free water, 1.25 μL of COVID-19 Real Time PCR Assay Multiplex, 6.25 μL of TaqPath™ 1-Step Multiplex Master Mix (No ROX™) (4×), and 5 µL template. 

The cDNA synthesis and amplification were performed with a CFX96 C1000 thermal cycler (Bio-Rad Laboratories, Philadelphia, PA, USA) following the recommended cycling conditions: reverse transcription at 50 °C for 20 min and denatured at 95 °C for 15 min, followed by 45 cycles of PCR at 95 °C for 10 s, 60 °C for 15 s, and 72 °C for 10 s. Results interpretation and Ct determination were performed with Applied Biosystems™ COVID-19 Interpretive Software 2.5 CE-IVD Edition. Targets detected with a Ct less than 40 were considered positive. A sample was considered positive if at least one of the targets showed an amplification signal. The sample amplification reaction was considered invalid if the internal control showed no amplification signal. 

### 2.6. Statistical Analyses

We estimated the sensitivity and specificity of gargle water LAMP with RT-PCR (of gargle water samples and of paired oro-nasopharyngeal swab samples) as the reference method. 

Sensitivity was calculated as the center value of the Wilson score interval of the proportion of positive LAMP results for gargle water samples, satisfying a given Ct value cut-off (with respect to their RT-PCR results or the RT-PCR results of their paired oro-nasopharyngeal swab samples) and, possibly, a given STT cut-off. Specificity was similarly calculated as the center value of the Wilson score interval of the proportion of negative LAMP results for gargle water samples with negative RT-PCR results or with paired oro-nasopharyngeal swab samples with negative RT-PCR results. Note that for all sensitivity and specificity calculations, the lower and upper bounds of the 95% Wilson score intervals were also calculated and stated.

For the final section of the data analysis, we studied the effect of STT on LAMP performance, while controlling for the sample Ct value. We did this by fitting a model to predict the LAMP result of a gargle water sample solely based on its RT-PCR Ct value, and then compared the predicted LAMP results of low and high STT samples against the empirically observed results. In more detail, a logistic model with L2 regularization was fitted, utilizing the RT-PCR results of gargle water samples as predictors for LAMP test results. Only gargle water samples with positive RT-PCR results were used when fitting the model, as negative RT-PCR results did not, by default, present numeric Ct values to perform the regression against. The goodness of fit of the logistic model was assessed via 10,000 Monte Carlo cross-validations in order to estimate its mean-accuracy and AUROC scores. After verifying a high goodness of fit, we exploited the logistic model as follows. The Ct values of gargle water samples with positive RT-PCR results (sub 40 Ct), which also satisfied STT ≤ 7 days, were inputted into the model to derive a set of conditional probabilities for observing positive LAMP results. Wald-based 95% confidence intervals [44] were computed for each individual term in this set of probabilities, yielding 95% confidence intervals for the probabilities of observing positive gargle water LAMP results for each individual Ct value that was inputted into the model. Thereafter, via the construction of Poisson binomial distributions, the probabilities outputted by the model were collectively compared against the actual empirical gargle water LAMP results of the relevant subsets of data, and two-tailed *p*-values were calculated for the empirical proportion of positive results, as seen in the empirical data. The previously derived 95% confidence intervals for the individual probabilities were then carried through the Poisson binomial distributions to yield a 95% confidence interval for the value of the two-tailed *p*-value. 

The model was then also applied, in the same manner, to derive a two-tailed *p*-value (with a 95% confidence interval for the estimate) for the probability of observing an equal or more extreme proportion of positive gargle water LAMP results than the proportion that was empirically observed for patients with STT ≥ 14 days. For a more detailed and technical overview of this analysis, please view Appendix A.

All data processing and analysis were performed in Python (ver. 3.9.2) and the modules used were “os”, “math”, “random”, “json”, “numpy” (ver. 1.20.1), “pandas” (ver. 1.2.2), “scipy” (ver. 1.6.1), “sklearn” (ver. 0.24.2), and “plotly” (ver. 4.14.3). Additionally, for the probability mass function of the Poisson binomial distribution, the module “poisson-binomial” (ver. 0.0.1) was used.

## 3. Results

### 3.1. Performance of Gargle Water LAMP with Respect to Paired Oro-Nasopharyngeal Swab and Gargle Water RT-PCR

In total, 202 oro-nasopharyngeal swab samples alongside 202 paired gargle water swab samples were collected. Out of the 202 gargle water samples, 72 yielded positive results on LAMP tests. In order to trace the performance of gargle water LAMP with respect to the gold standard RT-PCR method, we used, as seen in Table 1, 25, 30, and 40 as cut-offs for the Ct values delivered from the RT-PCR of paired oro-nasopharyngeal swab samples.

As the discrepancy of the LAMP and RT-PCR results in Table 1 could be partly attributed to the different sampling methods (gargle water for LAMP and oro-nasopharyngeal swabs for RT-PCR), Table 2 compares the LAMP results to the RT-PCR results of the same gargle water samples.

Furthermore, to supplement the gargle water LAMP sensitivity estimates (with respect to different Ct value cut-offs for paired oro-nasopharyngeal swab RT-PCR results) from Table 1, we present the relationship between continuously variable Ct cut-offs and the respective estimates for the sensitivity of LAMP in Figure 1. The figure features a 95% confidence interval band derived from connecting adjacent 95% Wilson score intervals.

### 3.2. Analysis of Gargle Water LAMP Results with Respect to STT 

Table 3 presents estimates for the sensitivity of gargle water LAMP with respect to gargle water and oro-nasopharyngeal RT-PCR results for sample pairs with low and high STT values. Note that low STT samples were defined as having STT ≤ 7 days (less than or equal to a week) and high STT samples were defined as having STT ≥ 14 (more than or equal to two weeks).

Additional data analysis was performed on gargle water LAMP results with respect to gargle water RT-PCR results for samples with either STT ≤ 7 days or STT ≥ 14 days. The aim of this analysis was to determine whether STT had an impact that was independent of a sample’s Ct value (taken as a proxy for viral load) on the rate of occurrence of positive gargle water LAMP results. 

Briefly, the strategy of the analysis consisted of the following:(1)Training a model to predict the LAMP result of a gargle water sample solely based on its RT-PCR derived Ct value and then verifying whether the model is accurate and appropriately fits the data.(2)Using the model to predict, via inputted Ct values, the expected proportion of positive LAMP results for samples satisfying STT ≤ 7 days and, separately, STT ≥ 14 days.(3)Comparing the expected proportions of positive LAMP tests against the empirically observed proportions and computing two-tailed *p*-values.(4)Accounting for the error of the model and computing 95% confidence intervals for the *p*-values calculated in the prior step.

Logistic regression was chosen as the model for the analysis (see Appendix A for details). From the ensuing analysis exploiting the model, we noted that the empirically observed proportion of “37/50” for the rate of positive LAMP results for gargle water samples with STT ≤ 7 (and positive gargle water RT-PCR results) was significantly greater than the expected proportion of “28.3/50”, even though the logistic model accounted for the Ct values of these samples, as, indeed the two-tailed p-value for observing a proportion as extreme as the empirically observed proportion, conditioned on the Ct values of these samples, was calculated to be p ≈ 0.004 (95% CI: p<0.001 to p ≈ 0.10). Furthermore, the proportion “4/24” for the samples with STT ≥ 14 was significantly lesser than the expected proportion (of “8.6/24”), as the two-tailed p-value for observing such an extreme proportion, conditioned on the Ct values of the STT ≥ 14 samples, was computed to be p ≈ 0.04 (95% CI: p ≈ 0.009 to p ≈ 0.16).

### 3.3. Sensitivity of Gargle Water Direct RT-PCR with Respect to Oro-Nasopharyngeal Indirect RT-PCR

Table 4 presents the results and sensitivity of gargle water direct RT-PCR with respect to oro-nasopharyngeal indirect RT-PCR at different Ct value cut-offs. By comparing Table 1 and Table 4, we note that for Ct value cut-offs of 25, 30, and 40, LAMP presents sensitivity estimates that are 21.5%, 15.2%, and 9.3% greater than the equivalent estimates for direct RT-PCR, respectively.

### 3.4. Comparison of RT-PCR for Gargle Water and Oro-Nasopharyngeal Swab Sample Pairs

In order to compare the impact of different sampling methods while controlling for effects due to different sample processing and amplification methods, we compared the RT-PCR of paired gargle water and oro-nasopharyngeal swabs. From the contingency table presented in Table 5, we note the agreement of the RT-PCR results for 163 out of the 202 sample pairs (80.7% agreement).

Furthermore, in Figure 2, the LAMP results and the distribution of Ct values for the 172 sample pairs are shown, where at least one of the two paired samples yielded a positive RT-PCR result.

## 4. Discussion

In our study, we performed LAMP on gargle water samples without any pre-enrichment steps, and we evaluated the suitability of this procedure for the detection of SARS-CoV-2 infections via comparison to gold-standard RT-PCR test results of paired oro-nasopharyngeal swab samples. We also assessed the impact of the assay type (direct RT-LAMP versus indirect RT-PCR) for SARS-CoV-2 detection by comparing the LAMP results to the RT-PCR results from the same gargle water samples. Furthermore, we assessed the impact of STT on LAMP sensitivity with respect to gargle water and oro-nasopharyngeal RT-PCR results. Furthermore, we exploited a logistic model to see if STT has an effect on LAMP performance, even when controlling for Ct values (taken as a proxy for viral load). We also compared the sensitivity of LAMP against another direct method, namely direct RT-PCR, performed on the same gargle water samples. Finally, we assessed the viability of gargle water sampling as a sampling method, by looking at the agreement of the indirect RT-PCR results of paired gargle water and oro-nasopharyngeal swab samples. 

The WHO’s priority target product profiles for COVID-19 diagnostics indicate a sensitivity of ≥ 80% and a specificity of ≥97% as key factors for determining an acceptable performance [45]. Recall that we principally assessed the sensitivity and specificity of LAMP with respect to the gold standard method (oro-nasopharyngeal RT-PCR). Gargle water LAMP appears to satisfy the WHO criteria for ≥80% sensitivity if, for a given gargle water sample, a paired oro-nasopharyngeal RT-PCR test would yield a sub-25 Ct result (Table 1).

Additionally, our results in Table 5 show that for sample pairs with STT ≤ 7 and where the oro-nasopharyngeal swab yielded a sub-30 Ct RT-PCR result, the sensitivity of the gargle water LAMP was estimated at 77.6% (95% CI: 66.2–89.1%), hence not satisfying, but approaching the WHO criterion for 80% sensitivity.

Note that the specificity of gargle water LAMP was estimated to be 93.9% (88.2–99.6%), corresponding to the proportion “43/44” (Table 1). That is, for the 44 sample pairs with negative oro-nasopharyngeal RT-PCR results, precisely one paired gargle water sample yielded a positive LAMP result. Therefore, our estimated specificity did not satisfy the WHO criterion of ≥97% specificity. Yet, the specificity of gargle water LAMP may be greater than 93.9% (and may potentially satisfy the WHO criterion) [16,46]. Indeed, we note that the relevant patient was admitted on the basis of COVID-19 symptoms 8 days prior to sampling and had an STT of 12 days at sampling. Hence, this patient may still have carried trace amounts of viral RNA. Furthermore, the gargle water LAMP test for this patient yielded, after amplification, not a yellow, but rather, an intermediate orange color, which may be regarded as a weakly positive result (i.e., indicating the possible presence of trace amounts of viral RNA in the sample). Following indications by the LAMP assay manufacturer [47] and conservative practice [48,49], orange was classified as positive in our trial, hence this false positive result. However, the flip side of regarding orange results as negative (thereby boosting specificity) would be a decreased test sensitivity; hence, from a clinical perspective, the case is not completely clear-cut. Therefore, in the end, a specificity higher than 97% can only be confirmed via further testing of the LAMP method on a greater sample size, potentially aiming to test individuals other than those presenting COVID-19 related symptoms (to provide for more true negatives). 

We further analyzed if the performance of the LAMP method is affected by the STT of gargle water samples independently of RT-PCR Ct values (a proxy for the viral load of the samples). This was done via a logistic model that predicts the LAMP result of a gargle water sample solely based off the Ct value derived from the RT-PCR of the sample. The model was then applied to the subsets of gargle water samples with positive RT-PCR results and STT ≤ 7 days (and separately to the subset of samples with STT ≥ 14 days). The empirically observed proportion of positive LAMP results in the group with STT ≤ 7 was significantly greater than that predicted by the model, and for the group with STT ≥ 14, the proportion was significantly lower than expected (see Results, Section 3.2). The unlikeliness of the empirical data for these low and high STT groups suggests that the relative performance of the direct LAMP method (with respect to the RT-PCR method) is affected, on average, by the STT value of the inputted samples independently of their Ct values, such that the method appears to be more sensitive for samples with lower STT values, even while keeping the underlying Ct values of the samples fixed.

Although, with increasing STT, samples may undergo biochemical changes [50,51,52] which negatively affect the viability of the LAMP method (e.g., greater amounts of RNA degradation or the appearance of sub-genomic variants), such changes should also impact the performance of the RT-PCR method, as both methods are nucleic acid amplification methods. Therefore, a more likely explanation may relate to RNA extraction, i.e., to the fact that we compared a direct LAMP method against an indirect RT-PCR method. Note that the term “indirect” refers to when a testing method was applied after RNA extraction. Here, “direct”, in turn, refers to when there only had been the heat lysis of a sample, as was the case for direct LAMP.

We propose that the higher effectiveness of gargle water direct LAMP at low STT values (e.g., STT ≤ 7), when assessed against an indirect method such as indirect RT-PCR, is attributable to hypothesized greater proportions of free RNA versus genomic RNA packed in virions for samples with low STT values (i.e., to a hypothesis that STT inversely correlates with the ratio of free RNA to genomic RNA packed in virions). In turn, the heat lysis in direct LAMP may not be lysing all of the present virions, and so direct LAMP, in comparison to indirect RT-PCR, would benefit from samples having high proportions of free RNA (i.e., having low STT values), while indirect RT-PCR, being an indirect method involving nucleic acid extraction, would not be as reliant on the presence of free RNA (i.e., on the sample being from low STT patients). 

The low stability of free RNA in the sample would further magnify this effect, as free RNA is directly exposed to RNAses and physical factors like freezing−thawing that may degrade it more rapidly. Indeed, a short sample storage-time may therefore be important for the performance of direct LAMP, and conversely, direct LAMP may be particularly effective when performed quickly after sampling, while the sample is still rich enough in free RNA (low STT or not). So, the hypothesized greater dependence of direct LAMP on free RNA (as opposed to indirect RT-PCR) would comply with the results, if indeed the proportion of free RNA (to genomic RNA in virions) is shown to be generally higher for samples with STT ≤ 7, while the relative proportion of virions is shown to be greater for a high STT (e.g., for STT ≥ 14). However, this matter was not investigated in our study, but may be a topic of interest for further investigation.

On another matter, our sensitivity estimates with respect to the gold standard for LAMP versus our equivalent estimates for direct RT-PCR suggest that out of these two direct methods, LAMP performs better, e.g., for a cut-off of 25 Ct (for the paired gold standard results), the sensitivity of direct RT-PCR was computed to be 59.7% (46.9–72.4%), whereas the equivalent estimate for LAMP was greater, being 80.8% (70.8–90.8%).

Furthermore, in this study, we assessed the viability of gargle water sampling in general by comparing the results of RT-PCR tests from paired gargle water and oro-nasopharyngeal swab samples. Firstly, we underline that the Ct values from the two samples types do not allow for a direct comparison of the actual viral loads present in the sampled oro-pharyngeal (in the case of gargle water sampling) and oro-nasopharyngeal regions (in the case of the swabs), as the oro-pharyngeal gargle water specimens, at sampling, underwent a high degree of dilution due to their mixing with the saline solutions used for gargling. Additionally, due to time and resource constraints under the pandemic, all of the gargle water samples, unlike all of the oro-nasopharyngeal swab samples, underwent a freezing and thawing cycle before sample processing, which may have affected the integrity of the RNA present in the gargle water samples [53]. Therefore, instead of a quantitative comparison of the Ct values derived from the RT-PCR results, we looked at the qualitative agreement of the results in terms of the classification of patients as positive or negative. Indeed, for 163 out of 202 sample pairs, the RT-PCR test results agreed, i.e., we observed a high percentage agreement of 80.7%. We also noted that even though 25 sample pairs delivered positive oro-nasopharyngeal RT-PCR results without correspondingly positive gargle-water RT-PCR results, the converse event also occurred for 14 sample pairs, indicating that it was not always clear-cut whether oro-nasopharyngeal swab sampling presented a superior route for the detection of SARS-CoV-2 infections or not. Hence, assuming that sufficient gargling takes place (5–10 s as specified under the study design), self-sampled gargle water presents acceptable levels of agreement with gold-standard oro-nasopharyngeal swab sampling, and so may be regarded as an alternative sampling method.

In summary, our results indicate that the gargle water LAMP test is not equivalent, in terms of sensitivity, to the gold standard oro-nasopharyngeal RT-PCR test. This is, presumably, due to two major factors: firstly, gargle water, while viable, may not be equivalent in terms of the captured viral load to oro-nasopharyngeal swab sampling; secondly, the lack of sample pre-processing in direct LAMP likely limits the performance of the LAMP assay. However, it is also due to these limiting factors that gargle water LAMP is convenient, simple, and potentially affordable enough to be utilized for mass screening applications [54], e.g., as an alternative to antigen tests, which, in a meta-analysis by Dinnes et al., presented sensitives ranging from 34.1% to 88.1%, depending on the brand and on the cohort of tested patients [29]. Indeed, if the LAMP procedure is further optimized (e.g., by including a lysis buffer), its performance may be superior to rapid antigen tests presenting unsatisfactory clinical sensitivities [26,27,28]. Moreover, in the context of mass-screening, it is worth noting that LAMP is very amendable to high-throughput testing [46,55].

Furthermore, in the meta-analysis by Dinnes et al., other rapid molecular assays demonstrated sensitivities ranging from 73% to 100% [29], which correspond to the sensitivities presented by gargle water LAMP, contingent on applying a 25 Ct value cut-off (with respect to paired gold standard oro-nasopharyngeal swab RT-PCR test results) or a cut-off of 30 Ct and an additional STT cut-off of 7 days or less. We reiterate that our analysis of the performance of direct LAMP with respect to STT suggested that the method performs particularly well if the STT of samples is low, e.g., STT ≤ 7 days. Moreover, we note that there is growing evidence to suggest that the majority of all infectious SARS-CoV-2 cases will be bounded by these Ct and STT cut-offs as well [30,36,37,56]. That is, the contagiousness of an individual who receives a positive oro-nasopharyngeal RT-PCR test with a high Ct value and presents a high STT value is believed to be low. Furthermore, we restate that it may be useful to avoid “over-diagnosing” high STT and high Ct value individuals who are not infectious. Hence, gargle water direct LAMP, especially if further optimized, may be considered as a testing option if resource and time-efficient screening needs to be performed to identify infectious individuals. 

## 5. Conclusions

Gargle water LAMP presents an acceptable sensitivity for patients with high viral loads, while still being resource and time efficient (and more comfortable than tests utilizing oro-nasopharyngeal swab sampling). In particular, the usage of gargle water LAMP tests may be particularly effective if symptomatic individuals are tested while their STT values are low, as our study has indicated that gargle water LAMP appears to be particularly sensitive if STT is low. Furthermore, a comparison of the RT-PCR results of paired gargle water and oro-nasopharyngeal swab samples suggests that gargle water presents a viable sample collection route. In conclusion, we reiterate that gargle water direct RT-LAMP, especially if further optimized for direct application (e.g., by optimizing sample preparation), may be considered as a viable tool for rapid patient screening/stratification when immediate decisions about patient care must be made, and/or where RT-PCR tests cannot be delivered in a timely manner.

## Figures and Tables

**Figure 1 diagnostics-12-00775-f001:**
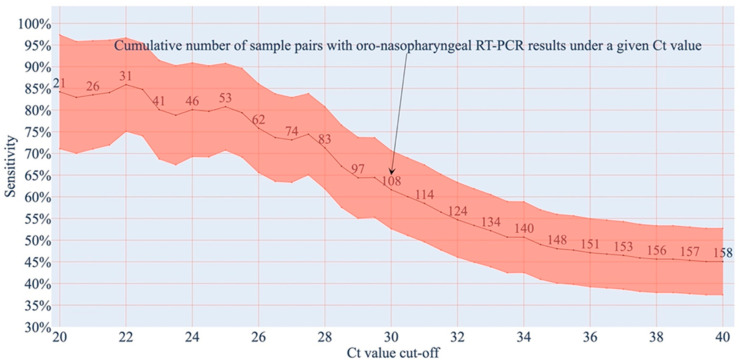
The cumulative sensitivity of LAMP displays a negative correlation with increasing Ct value cut-offs. The cut-offs were applied on the Ct values derived from the RT-PCR of paired oro-nasopharyngeal samples. The orange band presents 95% Wilson score (confidence) intervals for the sensitivities with respect to the given Ct value cut-offs. The black line presents the center values of these intervals. The numbers directly above the black center-value line indicate the number of sample pairs satisfying a given Ct value cut-off.

**Figure 2 diagnostics-12-00775-f002:**
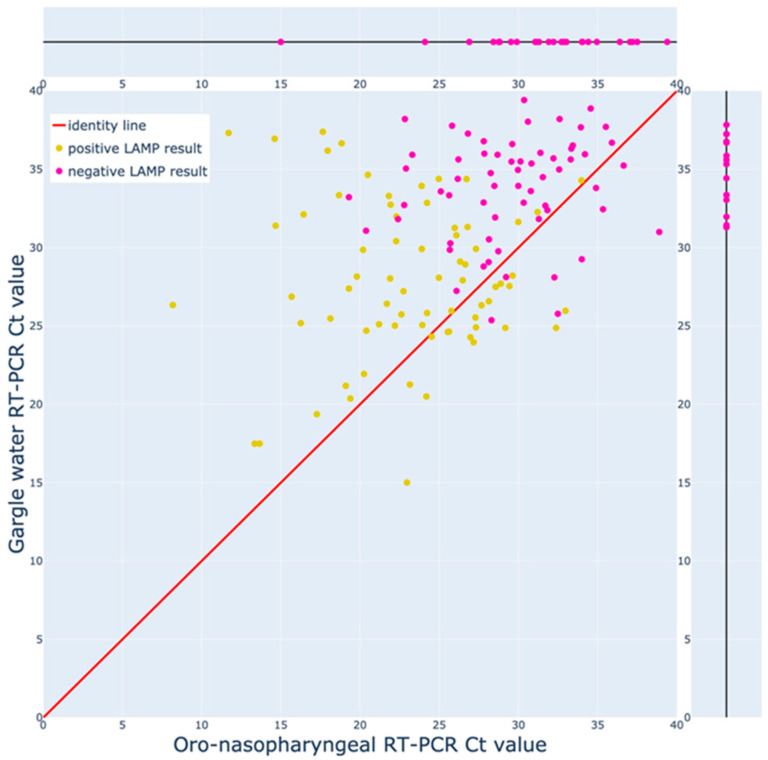
The RT-PCR Ct values and LAMP results for sample pairs with at least one positive RT-PCR result. The sample pairs where both the gargle water and swab yielded positive (sub 40 Ct) indirect RT-PCR results are distributed across the bottom left 2-D plane. For sample pairs where only one out of the two samples yielded a positive RT-PCR result, the corresponding data-point is situated on the 1-D line indicating its Ct value. For all data points, the color of the point indicates the gargle water LAMP result, with yellow corresponding to a positive result and purple to a negative result.

**Table 1 diagnostics-12-00775-t001:** The results of gargle water LAMP with respect to Ct value cut-offs for paired oro-nasopharyngeal RT-PCR results.

Ct Value Cut-Off for Oro-Nasopharyngeal RT-PCR Results	Ct ≤ 25		Ct ≤ 30		Ct ≤ 40	
	**Positive** **RT-PCR**	**Negative * RT-PCR**	**Positive** **RT-PCR**	**Negative * RT-PCR**	**Positive** **RT-PCR**	**Negative RT-PCR**
**Total**	53	149	108	94	158	44
**Positive LAMP**	44	28	67	5	71	1
**Negative LAMP**	9	121	41	89	87	43
	**Wilson score center value with 95% confidence interval**	**empirical** **proportion**	**Wilson score center value with 95% confidence interval**	**empirical** **proportion**	**Wilson score center value with 95% confidence interval**	**empirical proportion**
**Sensitivity**	80.8% (70.8–90.8%)	44/53	61.6% (52.6–70.6%)	67/108	45.1% (37.4–52.3%)	71/158
**Specificity**	**Not applicable ***				93.9% (88.2–99.6%)	43/44

* Note that for the 30 and 25 Ct value cut-offs, an oro-nasopharyngeal swab RT-PCR result was nominally regarded as negative if its Ct value was >30 and >25, respectively.

**Table 2 diagnostics-12-00775-t002:** The results of gargle water LAMP with respect to Ct value cut-offs for gargle water RT-PCR results.

Ct Value Cut-Off for Gargle Water RT-PCR Results	Ct ≤ 25		Ct ≤ 30		Ct ≤ 40	
	**Positive** **RT-PCR**	**Negative RT-PCR**	**Positive** **RT-PCR**	**Negative RT-PCR**	**Positive** **RT-PCR**	**Negative RT-PCR**
**Total**	18	184	58	144	147	55
**Positive LAMP**	18	54	48	24	71	1
**Negative LAMP**	0	130	10	120	76	54
	**Wilson score center value with 95% (confidence) interval**	**empirical proportion**	**Wilson score center value with 95% (confidence) interval**	**empirical proportion**	**Wilson score center value with 95% (confidence) interval**	**empirical proportion**
**Sensitivity**	91.2% (82.4–99.9%)	18/18	80.7% (71.1–90.4%)	48/58	48.3% (40.4–56.3%)	71/147
**Specificity**	**Not applicable ***				95.0% (90.4–99.7%)	54/55

* Estimating specificity with respect to the 25 and 30 Ct value cut-offs would entail regarding all the samples that had positive RT-PCR results above the given cut-offs as nominally negative. This would lead to artificially depressed specificity estimates and was not done. However, we did compute restricted sensitivity values, as our goal was to derive estimates of the sensitivity of gargle water LAMP when dealing with patients with strictly high viral loads.

**Table 3 diagnostics-12-00775-t003:** Gargle water LAMP sensitivity with respect to indirect RT-PCR and STT cut-offs.

RT-PCR Type:	RT-PCR of Paired Oro-Nasopharyngeal Samples		RT-PCR Result of Gargle Water Samples	
**STT Category**	**≤7 Days**	**≥14 Days**	**≤7 Days**	**≥14 Days**
**Ct cut-off:**	**Ct value ≤ 25**			
Proportion of samples with positive LAMP	29/32	3/5	9/9	3/3
Sensitivity estimate (95% Wilson score interval with center value)	86.3% (75.8–96.8%)	Insufficient sample size *	Insufficient sample size	Insufficient sample size
**Ct cut-off:**	**Ct value ≤ 30**			
Proportion of samples with positive LAMP	36/45	3/9	23/24	4/4
Sensitivity estimate (95% Wilson score interval with center value)	77.6% (66.2–89.1%)	Insufficient sample size	89.5% (79.8–99.3%)	Insufficient sample size
**Ct cut-off:**	**Ct value ≤ 40**			
Proportion of samples with positive LAMP	37/54	4/29	37/50	4/24
Sensitivity estimate (95% Wilson score interval with center value)	67.3% (55.3–79.3%)	18.0% (5.5–30.6%)	72.3% (60.4–84.1%)	21.3% (6.7–35.9%)

* Sensitivity estimates and 95% confidence intervals were not provided when the size of the computed confidence interval exceeded 30%, as this was taken as an indication that the relevant sample size was too small to provide a meaningful bound for the sensitivity.

**Table 4 diagnostics-12-00775-t004:** Binary results and sensitivity of gargle water direct RT-PCR with respect to Ct value cut-offs for paired oro-nasopharyngeal indirect RT-PCR results.

Ct Value Cut-Off for Indirect RT-PCR Results	Ct ≤ 25		Ct ≤ 30		Ct ≤ 40	
	**Positive Indirect** **RT-PCR**	**Negative Indirect RT-PCR**	**Positive Indirect** **RT-PCR**	**Negative Indirect RT-PCR**	**Positive Indirect** **RT-PCR**	**Negative Indirect RT-PCR**
**Total**	53	149	108	94	158	44
**Positive direct RT-PCR**	32	24	50	6	56	0
**Negative direct RT-PCR**	21	125	58	88	102	44
	**Wilson score center value with 95% (confidence) interval**	**empirical proportion**	**Wilson score center value with 95% (confidence) interval**	**empirical proportion**	**Wilson score center value with 95% (confidence) interval**	**empirical proportion**
**Sensitivity**	59.7% (46.9–72.4%)	32/53	46.4% (37.2–55.7%)	50/108	35.8% (28.4–43.2%)	56/158

**Table 5 diagnostics-12-00775-t005:** The results of gargle water indirect RT-PCR with respect to paired oro-nasopharyngeal indirect RT-PCR results.

	Positive Oro-Nasopharyngeal RT-PCR	Negative Oro-Nasopharyngeal RT-PCR
**Total**	158	44
**Positive gargle-water RT-PCR**	133	14
**Negative gargle-water RT-PCR**	25	30

## Data Availability

The de-identified raw data, alongside the code that produces the figures and statistical values presented in this paper, may be found on https://github.com/pouya-neuro/gargle_water_LAMP_trial. The last time the data was accessed: 21 March 2022.

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
