# Peer review of "Self-Sampled Gargle Water Direct RT-LAMP as a Screening Method for the Detection of SARS-CoV-2 Infections"

_diagnostics, 2022, doi:10.3390/diagnostics12040775_

Round 1
Reviewer 1 Report
The paper "Self-sampled gargle water direct RT-LAMP as a screening method for the detection of SARS-CoV-2 infections" presents an alternative method of diagnosis of SARS-CoV-2 infection, based on the RT-LAMP method, where the research material was water after gargling. The experiment was carefully planned and the methodology was described exhaustively. The statistical analysis of the obtained results was also applied accordingly and the results were thoroughly discussed. The manuscript is, in my opinion, very interesting and worth publishing.
Author Response
Thank you very much for taking your time to review this paper. We are grateful for your review and for your feedback.
Reviewer 2 Report
Arbaciauskaite et al. report on an evaluation of a SARS-CoV-2-specific LAMP assay. It is a nice little work contributing to the available literature on this topic. Prior to publication, I have a few suggestions on how the work could be further improved.
- My most important point is quite nicely depicted in Figure 1. As impressively shown there, the sensitivity of the assessed LAMP assay is in a quite similar range as well-designed immune-chromatographic rapid tests. Although the LAMP assay is quite easy-to-perform, it is hardly easier than a traditional rapid test. So the question is: What is the advantage of SARS-CoV-2-LAMP compared to traditional rapid testing.
Minor points:
- Abstract: Terms like “sub-8 STT” should be explained at first use
- Abstract: Focusing on the imperfect agreement of 80.7%, it is suggested to present not only sensitivity but also specificity.
- Introduction, lines 51-55: The authors should mention that there also other options of visualizing LAMP reaction, e.g., automated real-time LAMP.
- Introduction, lines 79-18 and Discussion, lines 505-507: Regarding potential “overdiagnosing” in case of declining viral loads during later stages of the infection, it should also be mentioned that very low viral loads can also be present at the very beginning of the infection. Exactly this is the reason why highly sensitive have been developed for infection control.
- Methods, lines 148-149: The authors mention a centrifugation step of the gargle water samples. It would be worth to at least shortly discuss arising biosafety issues associated with such a laboratory procedure with non-inactivated sample material.
7.Methods, lines 161-162 and 186-187: For the used commercial LAMP and real-time PCR assay, sensitivity and specificity as provided by the manufacturer and/or previous publications should be mentioned in the methods section to facilitate the interpretation of the results for the readers.
- Methods, Line 173 “1 log of RNA concentration”): I assume that a decadic log step is meant, isn’t it?
- Methods, lines 217-237: The authors may present a few explanatory sentences in the discussion sentence why they feel that such a model provides additional interpretative value instead to presenting descriptive statistics based on the raw data.
- Results, tables 1 and 2: I am a little bit uncertain about the interpretation of the p PPV and NPV values. The authors report that they wanted to evaluate the assay for screening purpose. In the assessment, however, the worked with samples specifically selected for a high-pretest probability hardly reflecting the situation in case of a screening. In case of a lower prevalence rate in the sample collection, however, PPV and NPV values would have been considerably different according to Bayes’ theorem. My suggestion is to either remove PPV and NPV or, alternatively, to explain this issue and how to interpret it in the discussion.
- Results, table 3: Obviously, the study was not powered for this assessment, as many fields remained open due to an insufficient sample size. Accordingly, it is doubtful whether this table really provides reliable information for this reason. The authors may want to comment on this.
- Results, lines 307 to 324: As already stated in my 9th point, the authors may want to explain why they feel that this sophisticated mathematical approach is really necessary to grasp the main findings of their study. The paper is not only read by mathematics experts and so, respective explanations seem advisable for readers less familiar with the mathematical approach.
- Results: Table 4 requires further explanation as well. CE-IVD accredited diagnostic assays are usually only valid in case of the application of an extraction approach included in this validation. Accordingly, changing the extraction procedure, which is part of the diagnostic algorithm, changes the complete test. So, it is difficult to see why the authors performed this approach not recommended by the manufacturer?
- Results, table 5: It would be interesting to learn about the Ct-value range of the sample with discordant test results. I assume it should be the higher Ct-value range, isn’t it?
- Discussion, line 388: “reasons to believe” is a pretty poor degree of scientifical evidence.
- Discussion, lines 396-397: If the authors state that reading “indeterminate” results as “positive” spoiled the assays specificity, they should also mention in how far generally reading them as “negative” would have further spoiled sensitivity. Otherwise, such an interpretation would be biased.
- Discussion, lines 428-437: We authors hypothesis on insufficient lysis might have been checked using electron microscopy. I agree that such an approach might have been beyond the scope of the study, however, the authors might want to mention the lacking assessment in a limitations paragraph of their study.
- Altogether, the discussion is pretty long a – about wide ranges – poorly covered with references. I suggest shortening the discussion by focusing on the discussion on the actual results of the study without too much emphasis on hypothetical speculations not proven by the data provided.
- Discussion, lines 489-494: Coming back to my first comment, this might be a good place to explain why LAMP should be use if traditional immune-chromatographic rapid tests reach similar performance. Also, it might be worth saying some words about benchmarking of the technique, e.g., regarding high-throughput assessments with high numbers of samples. To my experience, LAMP (other than LAMP-NGS-combinations) is more suited for low to moderate sample counts, but the authors may correct me if I am wrong with this impression.
- Discussion, figure 3: I don’t believe in mixing science with politics. As the authors may know, countries like UK have recently decided to end isolation even for SARS-CoV-2-positive individuals. Such decisions reflect public health policy and need for quarantine or isolation is – in fact – more a political decision rather than a logical consequence of a diagnostic test results. The authors may want to comment on this or – alternatively – simply remove the public health consequences as an obligatory “must” from the figure.
Author Response
Thank you very much for taking your time to review this paper. We are grateful for your valuable review and for your detailed and rich feedback. We have tried to address all of your points and have updated our manuscript accordingly.
- In the introduction and discussion of our updated paper (see lines 079-084 and 491-504 in the updated manuscript), we have clarified some points about the LAMP testing method which we performed in this study. We stress that this study is to be regarded as more of a proof-of-concept for the application of gargle water direct LAMP for screening in the clinical setting; e.g. we highlight that the assay was not optimized for direct usage* (and yet we did not perform RNA extraction / sample preprocessing in order to make the testing procedure more rapid and affordable) - and so we note that the performance of a finalized commercial LAMP assay may still be improved by optimizing it (e.g. by introducing a lysis buffer) for direct/rapid testing. Furthermore, we note the increasing evidence questioning the variable performance of commercial rapid antigen tests (RAT). Although we acknowledge that RAT can be a good choice for affordably and quickly identifying infections in their most acute stages, we contend, and more explicitly lay out in the updated manuscript, that gargle water direct LAMP, especially if the assay were to be optimized for direct usage, may serve as a convenient alternative to RAT, from usage in single use cases to high-throughput testing (to which it can easily be extended).
*https://international.neb.com/-/media/nebus/files/manuals/manuale2019.pdf?rev=640485747af74
Minor points:
- The term “sub-8 STT” was removed from the paper and replaced with “STT <= 7 days”.
- In the abstract it may not be clear that with the term “agreement”, we are not referring to sensitivity, but to the proportion of RT-PCR result pairs which agreed with each other. We did not deem it appropriate to frame the comparison of gargle water RT-PCR with oro-nasopharyngeal swab RT-PCR in terms of sensitivity and specificity, as both procedures use the same indirect RT-PCR procedure and so a positive result for one method and a negative result for the other does not imply that we should regard the former result as a false positive or the latter as a false negative (hence the aversion towards sensitivity and specificity).
- This point has been addressed (see lines 051-054 in the updated manuscript).
- This is a very valid point, but without further symptom and exposure information, this issue, of course, would apply to all forms of testing and not uniquely to LAMP - e.g. a very recent or dated SARS-CoV-2 infection may both lead to a RT-PCR result with a +35 Ct value. Nonetheless, we have now better addressed this point in the introduction about viral load and stage of infection under lines 087-090.
- In our updated manuscript we state, under lines 156-157, that every sample was handled in accordance with the standard biosafety rules (treated as possibly contagious).
- We added the manufacturer specifications for the LoD 95% of their products and the specificity information.
- Correct. In our updated manuscript this has been noted.
- This matter has been addressed in the updated statistical analyses, results and discussion (See lines 231-235, 314-329, 419-427, 525-526). In short, this model analysis procedure allows the effect of STT, independently of its correlation with Ct values, to be assessed. That is, it allows us to control for viral load (granted that Ct values are taken as a proxy for viral load) and assess the following: does STT independently impact the performance of LAMP? We highlighted this point about the purpose of the model and deferred more of the mathematical details to the addendum in the updated manuscript.
- As per your suggestion, PPV and NPV have been entirely removed from the paper due to their dependence on the disease prevalence rate.
- As you point out, estimates with particularly large confidence intervals were deemed unreliable due to insufficient sample sizes and were labeled as such. In all other cases, the Wilson score interval gives an estimate that reflects the small sample sizes via the size of the interval itself - so even for small sample sizes of 24-50 samples, reliable, albeit large, confidence intervals are produced. This follows as the coverage probability of the Wilson score method remains sufficiently close to the nominally stated (95%) value even at low sample sizes (granted the proportion is not too close to 0 or 1). In fact, the Wilson score method is explicitly recommended for the estimation of such intervals for sample sizes under 40:
https://repository.upenn.edu/cgi/viewcontent.cgi?article=1440&context=statistics_papers
- We recognize the concerns expressed here and we refer to our answer given to the 9th point.
- Indeed, the standard usage of CE-IVD accredited RT-PCR assays includes a RNA-extraction step. However, as seen in numerous other studies, it may be of interest to assess the performance of such assays under a direct procedure (some examples in references 40-42). In our study, direct RT-PCR was carried out to yield a comparator, which like LAMP, also employed a direct procedure, to help better evaluate the performance of direct LAMP.
- Correct - almost all such samples were in the higher Ct value range. Moreover, the Ct values of the samples with discordant results may be individually viewed right beneath table 5 in figure 2 (on the vertical and horizontal bars flanking the main graph). However, for the sake of brevity, and because a direct comparison of Ct values is problematic (e.g. gargle water samples were, by necessity, diluted) these statistics were not included.
- Tying in with the next point number 16, this section of the paper has been amended to better acknowledge the limitations of the study.
- Ideally, “indeterminate” patients which yield an orange result after sample amplification should be retested, as the patient has presented a weakly positive result (likely very low viral load) and so should not be ruled out from having an infection. For this reason, it may be sensible to tentatively regard such orange results as positive (following the recommendation of the manufacturer* and following references 48-49). However, as you correctly point out, entertaining the notion of treating orange results as negative (in order to argue that specificity is higher) is hypocritical if the flip side is not considered - namely the impact such a decision would have on sensitivity. We therefore now acknowledge this in the paper (see lines 408-415) and conclude that only a further study, specifically including non-COVID-19 subjects, can address the question about the specificity of gargle water LAMP.
*https://international.neb.com/faqs/2020/06/25/after-amplification-my-samples-turned-orange-rather-than-yellow-is-this-acceptable-how-do-i-interpret-the-results
- This is an interesting suggestion. Indeed, electron microscopy was out of our study and technical scope. Nevertheless, we will consider this proposal in the future studies.
- The discussion has been considerably condensed. In particular the more speculative sections have been shortened.
- We refer to the answer given for point number 1 - in particular we point out that LAMP is also well suited for high-throughput testing (e.g. 96 well-plates).
- This is a good point - the table and the accompanying discussion have been removed.
Reviewer 3 Report
Your research topic is quite interesting. Using gargle water for SARS-COV-2 testing good to be a promising alternative to the standard nasopharyngeal sampling. However, the sensitivity of this approach, logically due to lower viral concentrations in gargle water, as well as individually inadequate sample collection - in some cases - could lead to higher rates of false negative results. This is a risk implied by you but only indirectly by comparing your approach with RT-PCR especially at STT ≤ 7 and low CT values. I suggest that this point is more clearly emphasized in your discussion and conclusions.
In addition, Figure 3 is somehow redundant and could be omitted entirely. In any case, I do not think that LAMP as a molecular I say good replace antigen based tests which reflect the presence of the virus in its infectious form. Please specify this point and amend accordingly.
Author Response
Thank you very much for taking your time to review this paper. We are grateful for your review and for your feedback.
We amended the paper to stress the need for proper gargling (5-10 seconds), as outlined in the methodology to address the point about “individually inadequate sample collection”. See lines 486-490 of the updated manuscript. Although we stress that swabs samples may also be of variable quality.
Furthermore, we explicitly stress now in the discussion (see lines 492-496 in the updated manuscript), that lower viral load in gargle water presents a potential limitation for the gargle water direct LAMP method, and that its inferior performance in comparison to oro-nasopharyngeal indirect RT-PCR can be partially attributed to this factor.
We also entirely removed Figure 3 as the figure, especially by now, is redundant.
Regarding your last comment, in the introduction and discussion of our paper (see lines: 082-084 and 500-504 in the updated manuscript), we have clarified some points about the LAMP testing method which we performed in this study. This study is to be regarded as more of a proof-of-concept for the application of gargle water direct LAMP for screening in the clinical setting; e.g. we highlight that the assay was not optimized for direct usage* (and yet we did not perform RNA extraction / sample preprocessing in order to make the testing procedure more rapid and affordable) - and so we note that the performance, in order to yield a finalized commercial LAMP assay, may be improved by optimizing (e.g. using a lysis-buffer) the assay for direct/rapid testing. Furthermore, we note the increasing evidence (referenced in the updated manuscript, under lines 496-504) questioning the variable performance of commercial rapid antigen tests (RAT). Although we acknowledge that RAT can be a good choice for cheaply and quickly identifying infections in their most acute stages, we contend, and more explicitly lay out in the updated manuscript, that gargle water direct LAMP, especially if the assay were to be optimized, should not be ruled out as an alternative to RAT under all circumstances.
Finally, as per your kind suggestion, we reviewed the manuscript for spelling and grammatical errors and fixed a number of these in the updated manuscript. Thank you again for your review.
*https://international.neb.com/-/media/nebus/files/manuals/manuale2019.pdf?rev=640485747af74